# Processing of Carob Kernels to Syrup by Ultrasound-Assisted Extraction

Maria Lisa Clodoveo [1,†], Pasquale Crupi [1,*,†], Marilena Muraglia [2] and Filomena Corbo [2]

1 Interdisciplinary Department of Medicine, University Aldo Moro Bari, 70125 Bari, Italy; marialisa.clodoveo@uniba.it
2 Department of Pharmacy-Drug Sciences, University Aldo Moro Bari, 70125 Bari, Italy; marilena.muraglia@uniba.it (M.M.); filomena.corbo@uniba.it (F.C.)
* Correspondence: pasquale.crupi@uniba.it; Tel.: +39-347-125-2849
† These authors contributed equally to this work.

**Abstract:** Carob syrup is one of the most important carob products, which can have applications in pastry and confectionery, as a fruit preservative, but also in the pharmaceutical field because of the antimicrobial activity due to its polyphenol content. Carob syrup is traditionally made through a very time-consuming process, involving solid–liquid extraction in boiling water and concentration at a high temperature (>100 °C), which potentially causes the degradation of the active compounds (i.e., procyanidins or flavonol glycosides). Therefore, in this work, an alternative and less drastic method based on ultrasound technology was proposed to produce carob syrup. Processing conditions (i.e., time, temperature, and liquid–solid ratio) influencing the extraction of total soluble solids (TSS) and total phenolic compounds (TPC) were optimized using a central composite design coupled to response surface methodology. Reliable mathematical models allowed us to predict the highest TSS (24 ± 2 °Brix) and TPC (1.7 ± 0.5 mg/mL) values that could be obtained at 15 min, 35 °C, and 2 mL/g. Finally, a different HPLC-DAD phenolic pattern was determined between syrups produced by traditional and ultrasound methods; epicatechin, 4-hydroxycoumaric acid, and ferulic acid were more concentrated in the former, while procyanidin $B_2$, myricitrin, and quercitrin were prevalent in the latter one.

**Keywords:** carob pods; Amele cv.; flavonols; tannins; green extraction

## 1. Introduction

The carob tree (*Ceratonia siliqua* L., a tree of the pea family Fabaceae) is widely diffuse in Mediterranean countries and, recently, has spread to some areas where the climate is similar, such as California, Mexico, South Africa, and India. The world carob production is about 315,000 tons per year, with Spain, Italy, Morocco, and Portugal being the main producers [1].

The carob fruit is constituted of pulp (90%), especially containing sugars (in particular sucrose, glucose, and fructose) and tannins (both hydrosoluble, derived from gallic acid, and condensed, derived from flavan-3-ols), together with low levels of proteins and fats. The remaining constituent is seeds (10%), principally composed of galactomannans but also other bioactive components, such as minerals and some flavonoids [2–4]. The seeds are required since they are a source of gum (locust bean gum), which is used as a growth medium, thickener, and food stabilizer [5]; moreover, germ flour derived from the seeds is suggested as a dietetic human food [6]. While, the pulp has few applications, such as in the production of chocolate and confectionery, or as a waste for animal feed, and thus it has low economic value [1]. However, recently, the pulp has deserved particular interest because of its aforementioned high content of polyphenols and tannins, which have been recognized as antioxidants and radical scavengers with potential health-promoting effects [4]. In particular, tannins can be considered an effective alternative to antibiotics.

Indeed, a lot of studies exist in the literature on the use of tannin-rich extracts in traditional medicinal preparations to treat various ailments, including bacterial infections [7,8].

Among the traditional carob products deriving from non-industrial food processing methods, the carob syrup deserves a mention because, in addition to its folk use for softening and conserving seasonal fruits (thanks to the high concentration of sugars, reducing the water activity value), or for the preparation of cakes, cookies, and homemade confectionery, it shows antimicrobial activity due to its polyphenol content [9]. Carob syrup is traditionally made by subjecting the cut or ground pods to a very time-consuming solid–liquid extraction in boiling water, with concentration by evaporation at a high temperature (>100 °C) [10]. Of course, the long processing time and high temperature used are the main drawbacks of the process, which can determine the degradation/oxidation of the active compounds [11].

Over the last decades, alternative and sustainable extraction techniques have spread due to their time-saving and environmentally friendly properties with low-cost production of high-quality phenolic extracts [12]. Among them, ultrasound-assisted extraction (UAE) is relevant for the ease of use and cost-effectiveness of the equipment [13]; it is much quicker than traditional methods, since the acoustic cavitations of power US (preferably at frequencies ranging from 18 to 40 kHz) provoke cell wall rupture, favoring the solvent access into the cell and consequently the mass transfer of the cell content [14]. The efficiency of the UAE is generally affected by several factors, such as the solvent–solid ratio, solvent type and concentration, particle size, extraction time, and temperature [15]. For instance, the selection of a suitable solvent in UAE depends on the selectivity and solubility of the solvent toward targeted compounds, but also the temperature, which can influence the solvent's physical parameters including vapor pressure and, consequently, the acoustic cavitation and threshold [16]. Therefore, it is crucial to apply a suitable strategy to optimize the extraction conditions. In this sense, two alternative optimization techniques are commonly chosen: one-factor-at-a-time experiments and response surface methodology (RSM). The former consists of changing just a factor at a time, while maintaining all the others at constant values and is a more time-consuming approach; the latter allows the simultaneous optimization of the single factors along with their possible interactions by providing a polynomial equation that fits the experimental data [17].

Although the UAE use for syrup processing could be hypothesized to reduce the production time and energy costs, and enhance the polyphenol content in the final product, to the best of our knowledge, no other reports exist in the literature about this issue. Therefore, this study aimed to optimize UAE conditions influencing the extraction of total soluble solids (TSS) and total phenolic compounds (TPC) for the alternative preparation of carob syrup from deseeded pods (Amele cv.) using a central composite design (CCD) coupled to the RSM tool. The main phenolic compounds were also identified and quantified through HPLC-DAD analysis.

## 2. Materials and Methods

### 2.1. Plant Materials

The trial was carried out in 2021, on mature carob pods (*Ceratonia siliqua*) of the Amele variety, collected from the same tree cultivated in the Apulia region (Italy). After washing, deseeding, and cutting them into small pieces (2–3 cm), the fresh pods were grounded to a fine powder (only in the case of UAE) by an IKA A11 basic homogenizer (IKA, WERKE GMBH & CO.KG, Staufen im Breisgau, Germany) for subsequent processing.

### 2.2. Chemicals

Formic acid, ethanol, Folin–Ciocalteu reagent, methanol, and HPLC-grade water and acetonitrile were supplied from Merk Life Science SRL (Milano, Italy). Chlorogenic acid was purchased from Phytolab (Aprilia, Italy), and gallic acid, ferulic acid, 4-hydroxycoumaric acid, procyanidin $B_1$, procyanidin $B_2$, quercitrin, myricitrin, (-)-epicatechin, and quercetin

were purchased from Merk Life Science SRL. (Milano, Italy) and used as HPLC reference standards.

### 2.3. Carob Syrup Obtained by a Traditional Method

A total of 250 g of cut carob pods were soaked with water (500 mL) for 48 h. The mixture was boiled for 10 min and left to cool. Then, the pods were pressed, separated from the liquid, and rinsed with fresh water. Finally, the recovered liquid (~ 450 mL) was filtered through cotton wool and concentrated by boiling (~ 60 min). The preparation was conducted in triplicates along with TSS, TPC, and HPLC-DAD polyphenolic profile determinations.

### 2.4. Carob Syrup Obtained by the Ultrasound-Assisted Extraction Process

An ultrasonic water bath (Elmasonic P 30H, Elma Schmidbauer GmbH, Singen, Germany), working in continuous mode at a fixed sonication power (100 W) and frequency (37 kHz), and equipped with sensors to monitor the time and temperature, was employed for the UAE carob syrup preparation. According to the selected extraction conditions, each experiment was performed on 2 g of carob powder, passed through a laboratory test sieve of 4 mm (Endecotts LTD, London, England) to obtain uniformly sized particles, and carefully weighed (EU-C1200, Gibertini SRL, Novate Milanese, Milano, Italy) into 50 mL capped centrifuge tubes.

After the ultrasound treatment, the extracts were centrifuged at $4000 \times g$ for 15 min at 5 °C in an EPPENDORF centrifuge 5810R (Hamburg, Germany), filtered through a 0.45 μm syringe cellulose filter, and analyzed to determine TSS and TPC; then, it was concentrated under vacuum at 45 °C. Furthermore, polyphenols were characterized by HPLC-DAD analysis on the optimized extract.

### 2.5. TSS and TPC Determination

TSS (expressed as °Brix) was measured by using a portable refractometer (ATAGO PR32 – Levanchimica, Bari, Italy). TPC was determined according to the method by Milani et al. [18]: 50 μL of diluted syrups were mixed with 50 μL of Folin–Ciocalteu reagent, 50 μL of methanol, and 250 μL of distilled water. In addition, 200 μL of sodium carbonate (20%) and 400 μL of distilled water were also added. After incubation for 90 min at 30 °C, the absorbance at 700 nm (Perkin Elmer, Lambda Bio 20, Boston, MA, USA) was recorded. The assay was performed in triplicate and the final results were expressed as mg of gallic acid equivalents (GAE) per mL of syrup.

### 2.6. HPLC-DAD Analysis

HPLC-DAD identification and quantitation of the phenolic profile were performed on the optimal extract determined by RSM, as well as the traditional carob syrup, both after dilution with distilled water (1:3 $v/v$). HPLC system, as well as the adopted chromatographic conditions, were identical to those previously reported [14].

Positions of the absorption maxima ($\lambda_{max}$), absorption spectra profile, and retention times (RT) were matched with those from pure standards to allow the compound identification. Then, quantification of polyphenols was made by using the calibration curves of gallic acid ($R^2$ = 0.9975; LOD = 0.094 μg/mL; LOQ = 0.313 μg/mL), caffeic acid ($R^2$ = 0.9956; LOD = 0.0094 μg/mL; LOQ = 0.0313 μg/mL), and myricitrin ($R^2$ = 0.9974; LOD = 0.094 μg/mL; LOQ = 0.313 μg/mL).

### 2.7. Experimental Design and Statistical Analyses

A 3-factor and 2-blocks standard CCD was performed to optimize the effect of the extraction temperature ($X_1$), extraction time ($X_2$), and liquid–solid ratio ($X_3$), on the production of carob syrup in terms of TSS and TPC. Twenty randomized experiments were conducted, consisting of eight cube points, six star points, and six replications at the center values to evaluate the pure error sum of squares and lack of fit test (Figure 1).

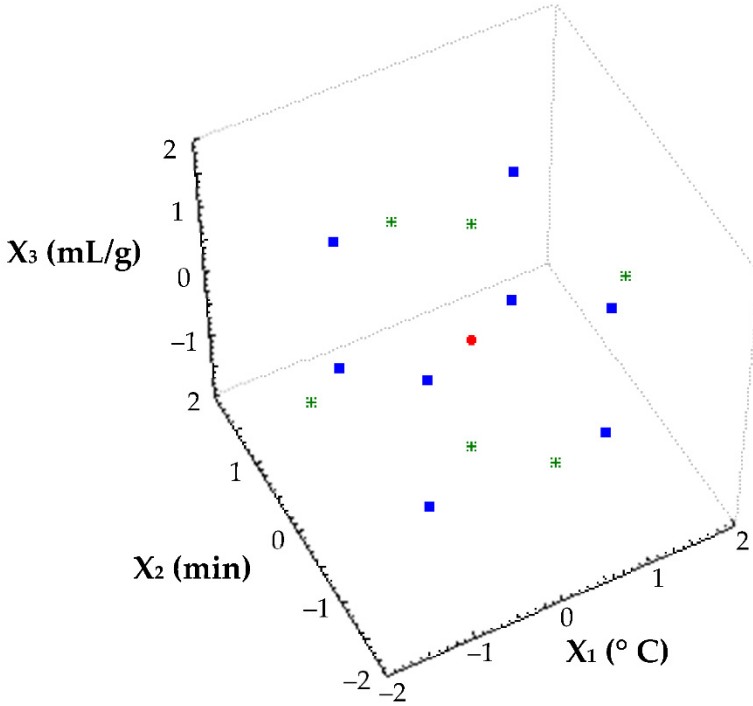

**Figure 1.** Standard CCD characteristics, eight cube points (blue squares), six star points (green crosses), and six center points (red circles).

CCD was designed and analyzed through RSM by using STATISTICA 12.0 (StatSoft Inc., Tulxa, OK, USA) software package. Two second-order polynomial equations (quadratic model) were developed to fit the TSS and TPC raw data experimentally obtained:

$$Y_i = B_0 + \Sigma B_i X_i + \Sigma B_{ii} X_i^2 + \Sigma B_{ij} X_i X_j$$

where Yi is the response function of each output variable; $B_0$ is a constant coefficient; $B_i$ are the regression coefficients of the linear, quadratic, and interactive terms, and $X_i$, and $X_j$ represent the coded independent variables ($X_1$, $X_2$, and $X_3$). According to the analysis of variance (ANOVA), the regression coefficients of individual linear, quadratic, and interaction terms were determined and the fitting of the mathematical models was employed by evaluating the $R^2$ and $R^2_{adj}$. Finally, further experimental extracts obtained under the optimized UAE were carried out for the model validation.

An independent *t*-test by groups was performed on the HPLC-DAD data from the analysis of the two carob syrups to eventually show the significant influence ($p < 0.05$) of the processing technique on the identified polyphenols.

### 3. Results and Discussion

*3.1. Optimization of UAE Parameters by CCD-RSM for Alternative Carob Syrup Preparation*

Figure 2 depicts the evolution of TSS (°Brix) measured at six time-points during the traditional preparation of carob syrup through soaking of deseeded and cut pods into water. As expected, a continuous increment of extracted sugars was observed in the range of 0–48 h, up to reaching the value of 24 ± 2 °Brix after boiling the solution for 10 min. Concerning the extracted polyphenols in this last stage, a mean value of 1.7 ± 0.5 mg GAE per mL of syrup was registered. Finally, after the separation from the solid part, the juice was concentrated by evaporation at a high temperature (>100 °C) until the commercial level of around 66 °Brix, to avoid reaching the super-saturation point [19].

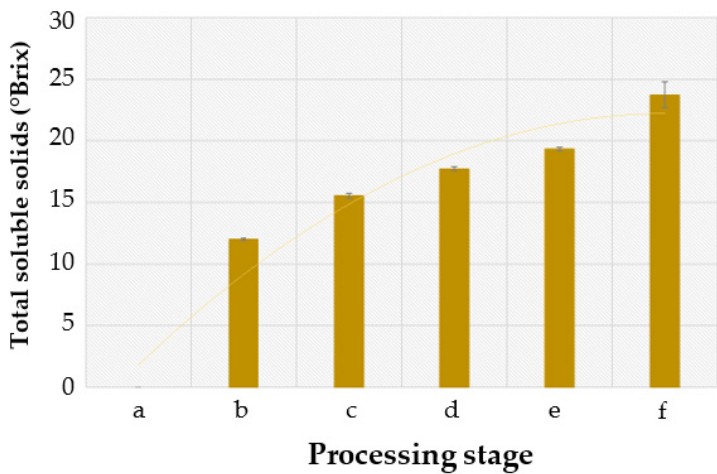

**Figure 2.** Evolution of TSS (°Brix) after (**a**) 0 h, (**b**) 12 h, (**c**) 24 h, (**d**) 36 h, (**e**) 48 h of the carob pods maceration in water, and (**f**) boiling for 10 min.

As described above, the traditional method to process the carob syrup was very time-consuming and potentially could provoke the degradation of the phenol chemical structure due to the high temperature used. Therefore, an alternative process was proposed consisting of the UAE of sugars and polyphenols from ground (particle size <4 mm) carob pods in water. A standard CCD (with $\alpha_O$ = 1.7638 and $\alpha_R$ = 1.6818 for orthogonality and rotatability, respectively) was employed to optimize the above selected three factors ($X_1$, $X_2$, and $X_3$), mostly affecting TSS and TPC in the carob syrup [20]. Table 1 reports the values of TSS and TPC, as well as the natural values of the factors for the 20 experiments, which were randomly executed to obtain an accurate estimation of the experimental error. To describe the empirical relationship between dependent variables and operational factors, the predictive second-order polynomial equations were generated by ANOVA analysis:

$$\text{TSS (°Brix)} = -0.67 - 0.008X_1^2 + 17X_3 - 4.75X_3^2 \quad (R^2 = 0.9434; R^2_{adj} = 0.9183) \quad (1)$$

$$\text{TPC (g/L)} = -2.33 - 0.009X_1^2 + 1.91X_3 - 0.41X_3^2 \quad (R^2 = 0.9408; R^2_{adj} = 0.9145) \quad (2)$$

**Table 1.** UAE optimization by three-level central composite design (CCD).

| Experiments | Block | $X_1$ (°C) | $X_2$ (min) | $X_3$ (mL/g) | TSS (°Brix) | TPC (mg/mL) |
|---|---|---|---|---|---|---|
| 8 | 1 | 45 | 17 | 2.5 | 18.0 | 0.998 |
| 3 | 1 | 25 | 17 | 1.5 | 20.0 | 1.042 |
| 1 | 1 | 25 | 8 | 1.5 | 22.2 | 0.869 |
| 4 | 1 | 25 | 17 | 2.5 | 18.0 | 0.833 |
| 11 | 2 | 53 | 13 | 2.0 | 17.2 | 0.749 |
| 9 (C) | 1 | 35 | 13 | 2.0 | 20.6 | 1.060 |
| 12 | 2 | 35 | 5 | 2.0 | 19.2 | 0.946 |
| 7 | 1 | 45 | 17 | 1.5 | 23.0 | 1.184 |
| 2 | 1 | 25 | 8 | 2.5 | 17.2 | 0.948 |
| 18 (C) | 2 | 35 | 13 | 2.0 | 18.0 | 0.952 |
| 13 | 2 | 35 | 21 | 2.0 | 19.4 | 0.887 |
| 20 (C) | 2 | 35 | 13 | 2.0 | 20.3 | 0.946 |
| 5 | 1 | 45 | 8 | 1.5 | 23.0 | 1.184 |
| 15 | 2 | 35 | 13 | 2.9 | 16.0 | 0.764 |
| 17 (C) | 1 | 35 | 13 | 2.0 | 22.0 | 1.522 |
| 6 | 1 | 45 | 8 | 2.5 | 17.0 | 0.752 |
| 14 | 2 | 35 | 13 | 1.1 | 14.0 | 0.474 |
| 10 | 2 | 17 | 13 | 2.0 | 17.0 | 0.617 |
| 16 (C) | 2 | 35 | 13 | 2.0 | 19.4 | 1.089 |
| 19 (C) | 1 | 35 | 13 | 2.0 | 19.8 | 1.163 |

　　　　The polynomials were reliable because the relative lack of fit test, performed on the six-fold repeated observations at the center point, as typically suggested in the case of three-factor CCDs [21], was not significant ($p > 0.05$). The determination coefficients ($R^2$) were both >0.9, meaning that <10% of the total variations were not explained by the models, as well as an overall good degree of correlation between the observed and predicted values. Then, the adjusted determination coefficients ($R^2_{adj}$) were close to $R^2$, confirming the statistical goodness of fit of the models.

　　　　The linear and quadratic terms of $X_3$ was the most significant factor influencing the extraction of both TSS and TPS, as demonstrated by the high values of regression coefficients associated with the linear and quadratic terms (see Equations (1) and (2)). When this factor ranged from 1 to 3 mL/g due to the volume variation, an increment of TSS and TPC were generally observed up to a maximum of around 2 mL/g, as illustrated by the response surfaces, which were generated based on the acquired polynomial equations (Figure 3). It is worth noting that this finding could be interpretable in disagreement with the literature reports, which state how an increase of the solvent–solid ratio generally results in better swelling of plant material, thus enhancing the mass transfer of soluble solids and polyphenols and, consequently, the yield of extraction [13,22]. Instead, as was also noted by Petit and Pinilla [20], the sugar extraction is more efficient as the water–kernel ratio increases, but at the same time, the sugar concentration measured decreases due to dilution.

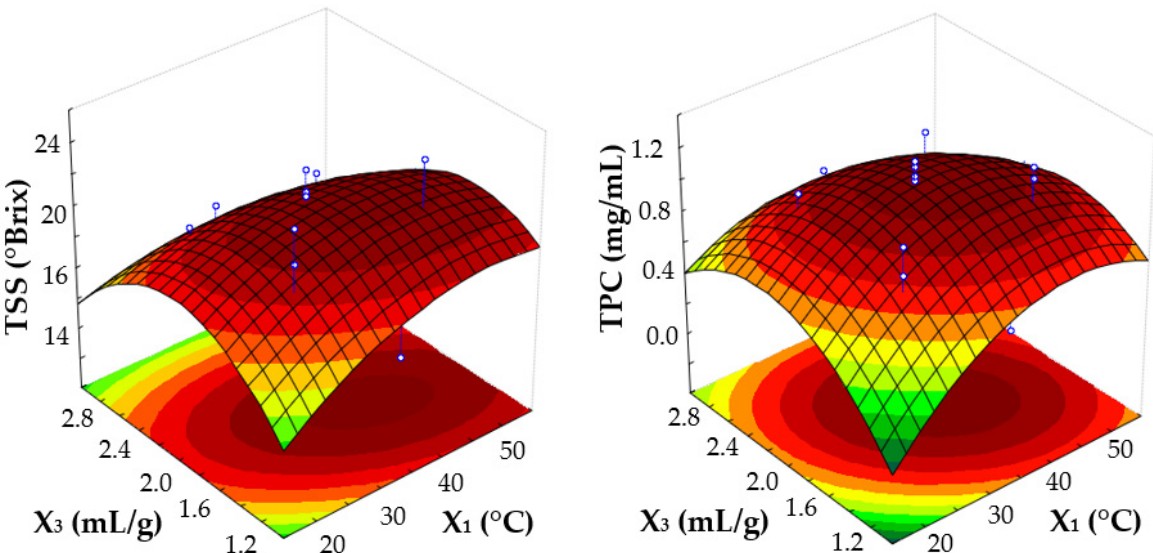

**Figure 3.** Response surface plots showing the effects of liquid–solid ratio ($X_3$) vs. temperature ($X_1$) on TSS and TPC in UAE carob syrup.

　　　　Furthermore, cavitation and implosion triggered by sonication lead to cell wall disruption which can justify an increased solvent permeation for an improved mass transfer and, thus, a higher extraction yield of sugars and polyphenols at this $X_3$ value, too [15].

　　　　Regarding the $X_1$ factor, its quadratic terms were just significant in the case of TSS and TPC, although the relative regression coefficients were very low (see Equations (1) and (2)). Anyway, sugars and total polyphenol concentrations, whose values initially increased upon raising the temperature (°C), reached a maximum level at around 35 °C, after which they started to decrease (Figure 3). Even though it was based on different extraction processes, our observation about the influence of temperature appeared similar to that of Petit and Pinilla, who found that the extraction efficiency was optimal up to 30 °C [20]. It is worth pointing out that as the extraction temperature increases, the viscosity and the surface tension of the solvent decrease, which brings an increase in vapor pressure. This increase in vapor pressure causes more solvent vapors to get inside the bubble cavity for cavitation,

which collapses less violently, hence it decreases the sonication effect. Moreover, at high temperatures, ultrasound can cause the formation of free radicals in the matrix, leading to the degradation of extractive compounds. Indeed, most studies stated the beneficial effect of a lower temperature of around 30 °C or below, in the case of UAE [16]. Then, concerning water in particular, the principal effect of the temperature increase is the weakening of its hydrogen bonds, reducing the dielectric constant; therefore, polar target compounds (i.e., sugars and some polyphenols), with high solubility in water at ambient conditions, are extracted most efficiently at lower temperatures [23].

Finally, no significant influence of extraction time ($X_2$) was registered; in the range of 5–20 min, TSS and TPC showed approximately the same amounts at the previously set $X_1$ and $X_3$ optimal values, reaching a plateau in their RSMs (Figure 4). However, an extraction time of 15 min was adopted for $X_2$ according to our choice in recent research dealing with the optimization of polyphenols ultrasound extraction from carob pods [14].

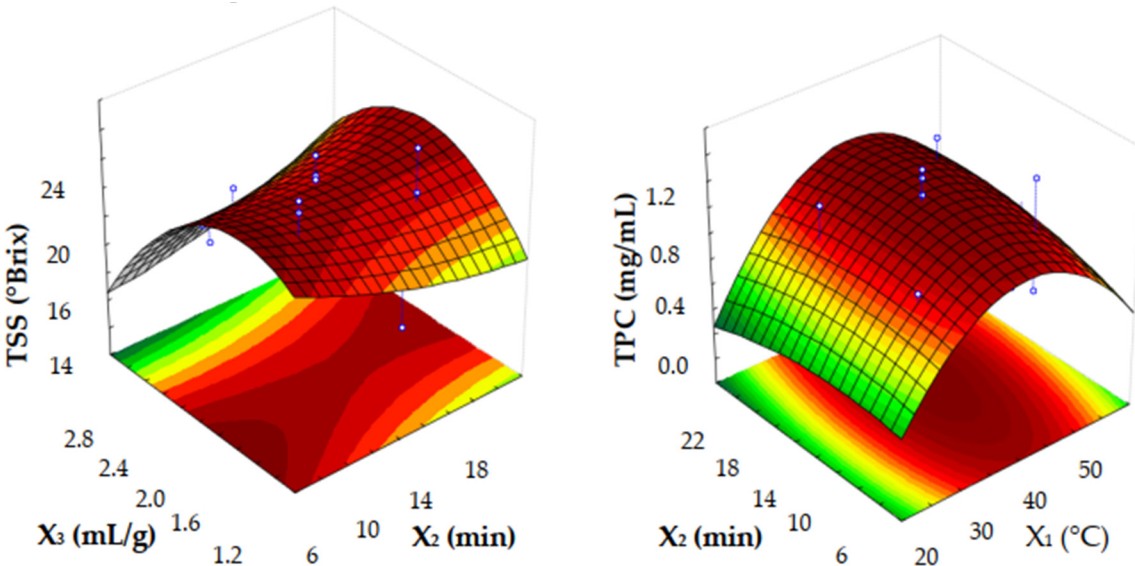

**Figure 4.** Response surface plots showing the effects of liquid–solid ratio ($X_3$) vs. time ($X_2$) and time ($X_2$) vs. temperature ($X_1$) on TSS and TPC, respectively, in UAE carob syrup.

The response optimization of UAE parameters for carob syrup preparation definitely led to the following experimental conditions: $X_1$ = 35 °C, $X_2$ = 15 min, and $X_3$ = 2 mL/g, from which TSS = 20.4 °Brix and TPC = 1.2 mg/mL were predicted. Therefore, to gauge the reliability of the predictive models, further extraction trials were conducted with the optimal conditions estimated by the RSMs and the gathered experimental data (TSS = 21.8 ± 1.6 °Brix and TPC = 1.1 ± 0.6 mg/mL) were in good agreement with the predicted values, confirming the effectiveness and validity of the RSM to establish the best UAE processing parameters.

### 3.2. HPLC-DAD Analysis of Polyphenols in Carob Syrup

Considering the similarity of TPC values between traditional and UAE syrup (as reported above), it could be inferred that the processing technology does not affect the content of phenolic compounds. Regardless, the UAE process is surely preferable due to using less solvent and lower power consumption, as well as the overall energy cost reduction (lower temperature and time than the traditional process), which would make the method suitable for industrial scale-up in response to the great demand for the standardization of carob products [10]. Anyway, TPC by Folin–Ciocalteu assay is recognized to only be a non-selective estimate of all reducents (including the polyphenols) present in a food matrix [24]. Moreover, polyphenols are sensitive to high temperatures in a way that strictly

depends on their structure [13]. For these reasons, the phenolic composition of the two syrups (diluted with distilled water, 1:3 *v/v*) was determined by HPLC-DAD analyses.

As described in a previous report [14], four benzoic and hydroxycinnamic acids, three condensed tannins, and two flavonoids were identified in both carob syrups by matching their retention times and UV absorption spectra with those of reference standards (Figure 5).

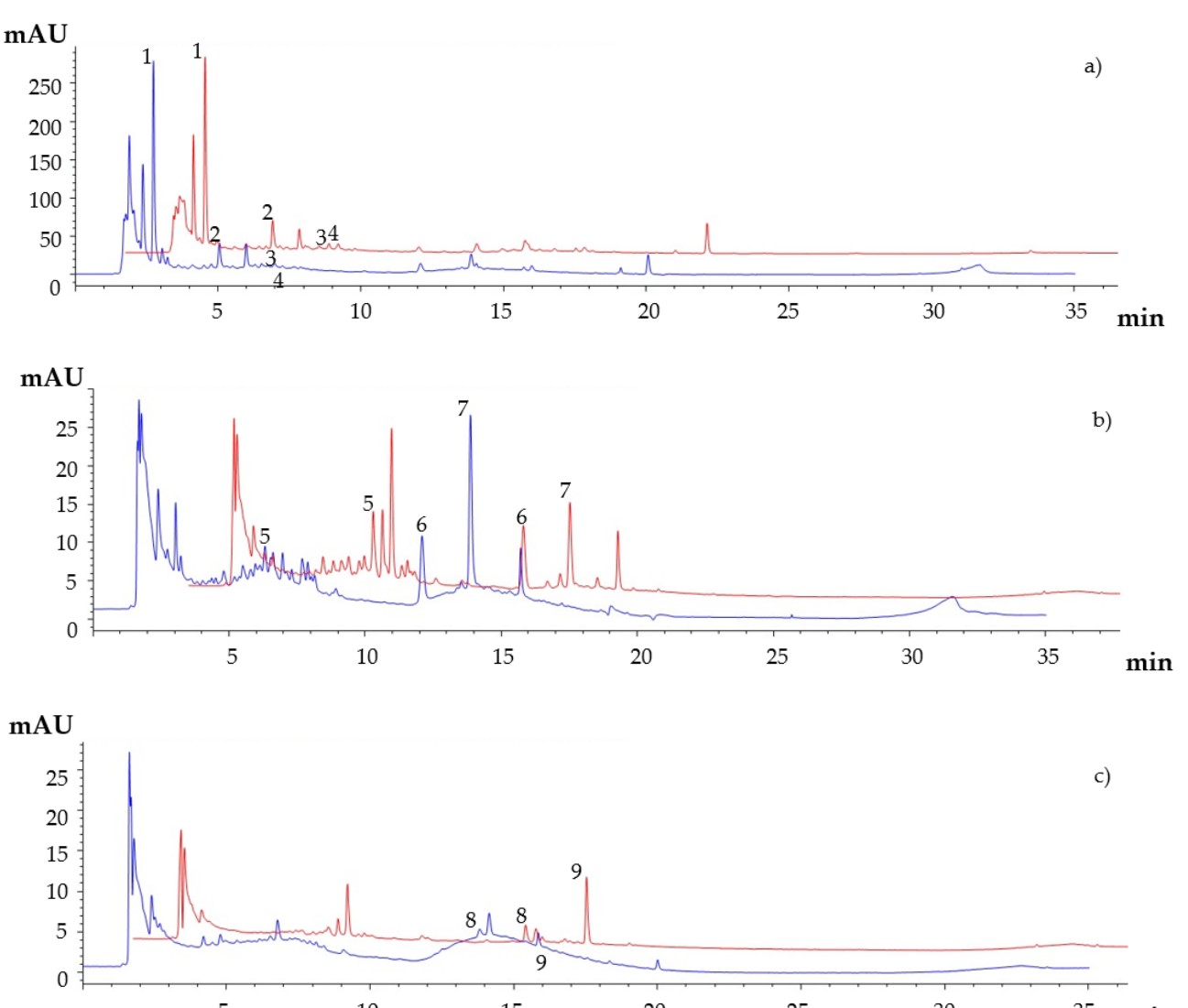

**Figure 5.** HPLC-DAD chromatograms at (**a**) 280 nm, (**b**) 330 nm, (**c**) 360 nm of traditional (blue line) and UAE (red line) carob syrup diluted 1:3. 1: gallic acid (2.733 min), 2: procyanidin $B_1$ (5.001 min), 3: procyanidin $B_2$ (7.091 min), 4: epicatechin (7.879 min), 5: chlorogenic acid (6.229 min), 6: 4-hydroxycoumaric acid (12.205 min), 7: ferulic acid (13.939 min), 8: myricitrin (13.531 min), 9: quercitrin (15.701 min).

The amounts of these compounds are listed in Table 2; from an independent *t*-test by groups; gallic acid, procyanidin $B_1$, and chlorogenic acid were not statistically different between traditional and UAE syrups. On the contrary, the other compounds were significantly ($p < 0.05$) influenced by the processing method; in particular, procyanidin $B_2$, myricitrin, and quercitrin were more concentrated in the UAE syrup, whilst epicatechin, 4-hydroxycoumaric acid, and ferulic acid were more concentrated in the syrup obtained by the traditional process (Table 2).

**Table 2.** Content of the identified phenolic compounds in traditional and UAE carob syrup.

| Compound | Traditional (µg/mL) | UAE (µg/mL) | Significance |
|---|---|---|---|
| Gallic acid | $124 \pm 19$ | $100 \pm 14$ | $p = 0.1555$ |
| Procyanidin $B_1$ | $18.7 \pm 1.5$ | $20.0 \pm 1.0$ | $p = 0.0788$ |
| Procyanidin $B_2$ | $2.70 \pm 0.08$ | $4.8 \pm 0.6$ | $p = 0.0049$ |
| Epicatechin | $1.11 \pm 0.19$ | $0.67 \pm 0.02$ | $p = 0.0153$ |
| Chlorogenic acid | $0.20 \pm 0.03$ | $0.26 \pm 0.04$ | $p = 0.1423$ |
| 4-hydroxycoumaric acid | $1.87 \pm 0.03$ | $1.76 \pm 0.08$ | $p = 0.0481$ |
| Ferulic acid | $3.35 \pm 0.08$ | $1.83 \pm 0.09$ | $p = 0.0026$ |
| Myricitrin | $1.3 \pm 0.4$ | $2.1 \pm 0.4$ | $p = 0.0436$ |
| Quercitrin | $2.1 \pm 0.6$ | $7.1 \pm 0.8$ | $p = 0.0011$ |

Experimental values are expressed as mean $\pm$ standard deviation of three replicates. The quantities of the identified polyphenols were compared by an independent *t*-test.

Various phenomena, including thermal stability, degradation due to the ultrasonic treatment, and mass transfer, could be invoked to explain this inhomogeneous behavior. For instance, it is acknowledged that flavonol glucosides like isoquercitrin and rutin, structurally similar to myricitrin and quercitrin, are very unstable at 70 °C and over [25]; in addition, procyanidin dimers can epimerize or cleave the interflavan linkage especially at 90 °C and in an aqueous medium [26]. Epicatechin can degrade either at high temperatures [27] or upon ultrasonic treatment [28], therefore its higher content in the traditional syrup (Table 2) may be ascribed to the aforementioned procyanidin $B_2$ (which is an epicatechin dimer) cleavage. Finally, regarding 4-hydroxycoumaric acid and ferulic acid, the finding is more complex to interpret because of their stability under ultrasound treatment in drastic conditions too [29] and their quick oxidation at high temperatures; therefore increased mass transfer coefficients coupled to an improved carob kernel wetting and penetration may be hypothesized [13]. However, further investigation is needed in this sense.

### 4. Conclusions

This research was aimed at optimizing the operating conditions (namely, solvent–solid ratio, time, and temperature), mainly affecting TSS and TPC levels, of an alternative process (UAE) to obtain carob syrup from the kernels of Amele cv. In this regard, a three-factor standard CCD coupled with RSM statistical tools were chosen. Linear and quadratic terms of solvent-to-solid ratio and temperature were the most significant factors influencing the process. Gathered results showed that the greatest TSS and TPC content could be obtained by ultrasound treatment at 2 mL/g, for 15 min, at 35 °C; moreover, their values could be predicted by reliable mathematical models, with determination coefficients higher than 0.9, built in to this study.

Then, from the comparison of syrups produced by the traditional method and UAE, it can be concluded that our starting hypothesis was partially verified; indeed, a similar content of TPC, produced in a much shorter time than the traditional method, was measured in the UAE syrup. Even though, the HPLC-DAD pattern of polyphenols was different between the two syrups, with procyanidin $B_2$, myricitrin, and quercitrin more concentrated in the UAE syrup, while epicatechin, 4-hydroxycoumaric acid, and ferulic acid were more concentrated in the traditional one.

In conclusion, the UAE process is surely preferable due to using less solvent and lower power consumption, as well as the overall energy cost reduction (lower temperature and time than the traditional process), which would enable an easier scaling up to industrial production of carob syrup, in response to the need for the standardization of carob products.

**Author Contributions:** Funding acquisition, project administration, data curation, writing—original draft, methodology: M.L.C.; project administration, data curation, writing—original draft, methodology: P.C.; writing—review & editing: M.M.; supervision: F.C. All authors have read and agreed to the published version of the manuscript.

**Funding:** This study was supported by grant from the Apulian Region (Research for Innovation REFIN-POR Puglia FESR-FSE 2014/2020 and "CE.SI.R.A.-CEratonia SIliqua Ri-sorsa genetica Autoc-tona da valorizzare" P.S.R. Puglia 2014/2020-Misura 16–Coopera-zione-Sottomisura 16.2 "Sostegno a progetti pilota e allo sviluppo di nuovi prodotti, pratiche, processi e tecnologie").

**Conflicts of Interest:** The authors declare no conflict of interest.

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
