# Peer review of "Processing of Carob Kernels to Syrup by Ultrasound-Assisted Extraction"

_processes, doi:10.3390/pr10050983_

Round 1
Reviewer 1 Report
Type of the Paper (Research Article)
PROCESSING OF CAROB KERNELS TO SYRUP BY ULTRASOUND-ASSISTED EXTRACTION
Comments to authors:
Line no 15: which active compounds are degraded? Mention
Line no 17: do these processing conditions affect the extraction positively?
Line no 36: which bioactive compounds? Discuss
Line no 46: only provide one of the updated references.
Line no 53: how carob syrup helps to preserve fruits? Discuss
Line no 59: give only updated references and avoid many references for one line.
- Please check this latest paper about extraction. Processes 9(8), 1406
Line no 68: solvent selection will be done on the basis of?
- Please check this latest paper about extraction. An inclusive overview of advanced thermal and nonthermal extraction techniques for bioactive compounds in foods and food-related matrices. Food Reviews International.
Line no 90. Scientific name should be italic.
Line no 223. TTS
Line no 228: UAE affects the sugars in which manner? Not discussed properly
Line no 282: graphical representation is appreciable.
Line no 334: if the polyphenol content is same like conventional which is the main component of carob syrup why ultrasound assisted extraction chosen?
Line no 384: Tables data management is commendable.
Line no 390: only their degradation gets affected or their percentage too?
Line no 405: conclusion is too short to get a whole idea of research try to write it in more comprehensible way.
- Overall the discussion of the paper is not very commendable. It’s too short and not properly linked.
- Objective of the research is accomplished? according to results poly phenol content remains unaffected. Justify please.
Author Response
Reviewer 1
Comments to authors:
Line no 15: which active compounds are degraded? Mention
According to your suggestion, some examples of polyphenols that are potentially degradable have been mentioned. “(i.e., procyanidins or flavonol glycosides)” – line 16, revised manuscript.
Line no 17: do these processing conditions affect the extraction positively?
We did not specify that because, generally, time, temperature, and liquid-solid ratio can influence the extraction either positively or negatively. It depends on their values as well as their interaction, which we aimed to optimize in this work. For example, in general, the increase in temperature leads to an increase in extraction yield; however, most studies stated the beneficial effect of the lower temperature of 30 °C or below in the case of UAE.
Line no 36: which bioactive compounds? Discuss
Accordingly, it has been modified as such: “…bioactive components, such as minerals and some flavonoids” (line 38, revised manuscript) and a relevant reference (Goulas, V.; Stylos, E.; Chatziathanasiadou, M.V.; Mavromoustakos, T.; Tzakos, A.G. Functional components of carob fruit: Linking the chemical and biological space. Int. J. Mol. Sci., 2016, 17(11), 1875) has been added.
Line no 46: only provide one of the updated references.
We decided to insert the following review: Goulas, V.; Stylos, E.; Chatziathanasiadou, M.V.; Mavromoustakos, T.; Tzakos, A.G. Functional components of carob fruit: Linking the chemical and biological space. Int. J. Mol. Sci., 2016, 17(11), 1875, which is a recent work dealing with all health benefits of carob.
Line no 53: how carob syrup helps to preserve fruits? Discuss
The carob syrup capacity of preserving fruits can be ascribed to the high concentration of sugars (> 65%), which contribute to reducing the water activity as well as the antimicrobial activity of polyphenols which block the proliferation of the microorganisms. Anyway, we would prefer to not face a deepened discussion on a theme that is beyond the aim of this article. Therefore, we have simply added the sentence “((thanks to the high concentration of sugars, reducing the water activity value)” lines 55-56.
Line no 59: give only updated references and avoid many references for one line.
According to your suggestion, the reference from Roseiro et al., 2013 has just been left.
Please check this latest paper about extraction. Processes 9(8), 1406
This review has been evaluated and opportunely cited in the text (line 73, revised manuscript).
Line no 68: solvent selection will be done on the basis of? Please check this latest paper about extraction. An inclusive overview of advanced thermal and nonthermal extraction techniques for bioactive compounds in foods and food-related matrices. Food Reviews International.
As required, some observation about solvent selection has been added as such: “For instance, the selection of a suitable solvent in UAE depends on the selectivity and solubility of the solvent toward targeted compounds, but also the temperature which can in-fluence the solvent physical parameters including vapor pressure and, consequently, the acoustic cavitation and threshold [16].” (Lines 73-77, revised manuscript).
Line no 90. Scientific name should be italic.
The name has been typed correctly in the revised manuscript (line 95, revised manuscript).
Line no 223. TTS
Correction done (line 243, revised manuscript).
Line no 228: UAE affects the sugars in which manner? Not discussed properly
As requested, the discussion on the possible effects of US on sugars and also polyphenols extraction has been extended. “Furthermore, cavitation and implosion triggered by sonication lead to cell-wall disruption which can justify an increased solvent permeation for an improved mass transfer and, thus, a higher extraction yield of sugars and polyphenols at this X3 value, too [15].” (lines 307-309, revised manuscript) and “It is worth pointing out that as the extraction temperature increases, the viscosity and the surface tension of the solvent decrease, which brings an increase in vapor pressure. This increase in vapor pressure causes more solvent vapors to get inside the bubble cavity for cavitation, which collapses less violently hence decreasing the sonication effect. Moreover, at high temperatures, ultrasound can cause the formation of free radicals in the matrix, leading to the degradation of extractive compounds. Indeed, most studies stated the beneficial effect of a lower temperature of around 30 °C or below in the case of UAE [16]. Then, concerning water, in particular, the most important effect of the temperature increase is undoubtedly the weakening of its hydrogen bends, resulting in a lower dielectric constant; hence polar target compounds (i.e., sugars and some polyphenols) with high solubility in water at ambient conditions are extracted most efficiently at lower temperatures [23].” (lines 316-328, revised manuscript).
Line no 282: graphical representation is appreciable.
Thank you very much for your comment.
Line no 334: if the polyphenol content is same like conventional which is the main component of carob syrup why ultrasound assisted extraction chosen?
Of course, you are right about the similarity of the two processes on the content of TPC. However, UAE process is surely preferable due to the less solvent and power consumption as well as the overall energy costs reduction (lower temperature and time than the traditional process). Moreover, the flow-mode extraction processes using cavitation phenomena enable an easier scaling up to industrial production, therefore narrowing the general gap between research laboratories and industry and, in particular, responding to the need for the standardization of carob products. These concepts were opportunely reported in lines 368-372 and in the conclusion section.
Line no 384: Tables data management is commendable.
Thank you very much for your comment.
Line no 390: only their degradation gets affected or their percentage too?
We are not sure to understand your issue. Anyway, we were interested to demonstrate that the higher temperature involved in the traditional process actually affects the concentration of polyphenols, depending on their structure. Of course, this also implied a variation in their percentages.
Line no 405: conclusion is too short to get a whole idea of research try to write it in more comprehensible way.
We think we have now extended the conclusion in a more comprehensive way (lines 446-464, revised manuscript).
Overall the discussion of the paper is not very commendable. It’s too short and not properly linked.
According to your useful suggestions, we have carefully extended the discussion.
Objective of the research is accomplished? according to results polyphenol content remains unaffected. Justify please.
Our hypothesis in this work was to reduce the production time and energy costs, and enhance polyphenols content in the final product, by using UAE for carob syrup processing in alternative to the traditional method. As stated in conclusion, this objective was partially accomplished because similar content of polyphenols as the traditional process was obtained but in a very shorter time and with less solvent and low temperature. Moreover, particularly health-promoting compounds such as procyanidins, myricitrin, and quercitrin were more recovered by UAE.

Reviewer 2 Report
The manuscript Processing of carob kernels to syrup by ultra-2 sound-assisted extraction submitted to Processes - Manuscript Number: processes-1713826 describes an alternative method of producing carob syrup. The article is original and well structured. I do not find important mistakes. Consequently, I recommend the manuscript for publication.
Author Response
Reviewer 2.
The manuscript Processing of carob kernels to syrup by ultra-2 sound-assisted extraction submitted to Processes - Manuscript Number: processes-1713826 describes an alternative method of producing carob syrup. The article is original and well structured. I do not find important mistakes. Consequently, I recommend the manuscript for publication.
Thank you very much for your approval.

Reviewer 3 Report
Authors evaluate the possibility of processing the carob pods into a syrup using ultrasound assisted extraction instead of a traditionally process and observe the effect on total solids and total phenolic content. The subject of the research is interesting and novel as there are no similar reports.
Firstly, authors are advised to carefully check the English language as there are many grammatical errors. Additionally, authors should carefully read the journal instructions for references as the style they applied is not proposed by the journal (authors use both numbers and authors names with year of publication). Regarding the references, in line 127 add the number of reference.
In methodology section, when describing the HPLC method, please clarify if it was adopted from previous study (and reference it) or it is newly developed method.
Generally, apart from the referencing and English, I would not have any concerns regarding this manuscript, but one issue raises doubts about this research. As the author state, commercial carob syrup contains 66 °Brix. By traditional method, authors soaked and then boiled the carob pods for 10 min and achieved 24 °Brix, and afterwards evaporated it to 66 °Brix by boiling (for how long?). With optimized UAE, pods were treated for 15 min and obtained “syrup“ had 21 °Brix. Authors state that this solution was filtered and analyzed as comparison to traditionally obtained syrup. I must emphasize that these two are not comparable, as the solution obtained by UAE is not syrup?! In that case UAE served only as a pretreatment for the extraction and should be followed by concentration, or otherwise traditionally prepared syrup should be analyzed before concentration and those two compared as pretreatment solutions. Thus, I must demand major revision from authors and clear description of the process, subject (syrup or semi-finished product?) and consequently the discussion of the research.
Author Response
Reviewer 3.
Authors evaluate the possibility of processing the carob pods into a syrup using ultrasound assisted extraction instead of a traditionally process and observe the effect on total solids and total phenolic content. The subject of the research is interesting and novel as there are no similar reports.
Thank you very much for the positive judgment.
Firstly, authors are advised to carefully check the English language as there are many grammatical errors. Additionally, authors should carefully read the journal instructions for references as the style they applied is not proposed by the journal (authors use both numbers and authors names with year of publication). Regarding the references, in line 127 add the number of reference.
According to your suggestion, the English language and style have been carefully revised.
We are very sorry about the mistake in the references style in the text; we have opportunely corrected that in the revised manuscript.
The number of the reference has been added in the new version of the manuscript (line 133).
In methodology section, when describing the HPLC method, please clarify if it was adopted from previous study (and reference it) or it is newly developed method.
Actually, the HPLC method was previously reported, but in this work, it was adapted to a different matrix. Anyway, the right reference has been added (line 143, revised manuscript).
Generally, apart from the referencing and English, I would not have any concerns regarding this manuscript, but one issue raises doubts about this research. As the author state, commercial carob syrup contains 66 °Brix. By traditional method, authors soaked and then boiled the carob pods for 10 min and achieved 24 °Brix, and afterwards evaporated it to 66 °Brix by boiling (for how long?). With optimized UAE, pods were treated for 15 min and obtained “syrup“ had 21 °Brix. Authors state that this solution was filtered and analyzed as comparison to traditionally obtained syrup. I must emphasize that these two are not comparable, as the solution obtained by UAE is not syrup?! In that case UAE served only as a pretreatment for the extraction and should be followed by concentration, or otherwise traditionally prepared syrup should be analyzed before concentration and those two compared as pretreatment solutions. Thus, I must demand major revision from authors and clear description of the process, subject (syrup or semi-finished product?) and consequently the discussion of the research.
Yes, your observation is right because in the previous form of the manuscript the solution obtained by UAE could not represent a syrup. Actually, we forgot to specify in M&M that the solution was finally “concentrated under vacuum at 45 °C” (see line 128, revised manuscript) until it reached the same TSS (~ 66 °Brix) as the traditional syrup. Indeed, as already reported in the submitted manuscript the phenolic composition of the two syrups was determined by HPLC-DAD analyses after dilution with distilled water, 1:3 v/v.

Round 2
Reviewer 1 Report
Great work
Wish you good luck.
Reviewer 3 Report
Authors have adressed all comments, so I suggest the manuscript for publication.